# A Systemic Insight into Exohedral Actinides and Endohedral Borospherenes: An&B_m_ and An@B_n_ (An=U, Np, Pu; m = 28, 32, 34, 36, 38, 40; n = 36, 38, 40)

**DOI:** 10.3390/molecules27186047

**Published:** 2022-09-16

**Authors:** Peng Li, Jingbo Wei, Hao Wei, Kerong Wang, Jizhou Wu, Yuqing Li, Wenliang Liu, Yongming Fu, Feng Xie, Jie Ma

**Affiliations:** 1School of Physics and Electronics Engineering, State Key Laboratory of Quantum Optics and Quantum Optics Devices, Institute of Laser Spectroscopy, Shanxi University, Taiyuan 030006, China; 2Collaborative Innovation Center of Extreme Optics, Shanxi University, Taiyuan 030006, China; 3Collaborative Innovation Center of Advanced Nuclear Energy Technology, Key Laboratory of Advanced Reactor Engineering and Safety of Ministry of Education, Institute of Nuclear and New Energy Technology, Tsinghua University, Beijing 100084, China

**Keywords:** actinides, borospherenes, bonding characteristic, density functional calculations

## Abstract

A series of exohedral actinide borospherenes, An&B_m_, and endohedral borospherenes, An@B_n_ (An=U, Np, Pu; m = 28, 32, 34, 36, 38, 40; n = 36, 38, 40), have been characterized by density functional theory calculations. The electronic structures, chemical bond topological properties and spectra have been systematically investigated. It was found that An@B_n_ is more stable than An&B_n_ in terms of structure and energy, and UB_36_ in an aqueous solution is the most stable molecular in this research. The IR and UV-vis spectra of An&B_m_ and An@B_n_ are computationally predicted to facilitate further experimental investigations. Charge-transfer spectroscopy decomposes the total UV-Vis absorption curve into the contributions of different excitation features, allowing insight into what form of electronic excitation the UV–Vis absorption peak is from the perspective of charge transfer between the An atoms and borospherenes.

## 1. Introduction

Metal-doped borospherenes have generated a lot of interest in the scientific community and massive amounts of research have been performed regarding endohedral and exohedral borospherene. Boron can form a variety of different compounds, including boron nanotubes [1,2], planar or quasi-planar structures [3,4,5,6,7,8,9], borospherenes and core-cell structures [10,11], due to its short covalent radius, lack of electrons, and high coordination number.

In 2014, Zhao et al. synthesized the first all-borofullerene [12], which set off a wave of experimental and theoretical research into borospheres. In the same year, Jian et al. reported the prediction of a B_38_ [13] fullerene analogue with the first-principles calculation. In 2015, Zhao et al. theoretically predicted the smallest all-boron cage, B_28_, composed of two B_12_ units [14]. Its structure is superior to other isomers (bicyclic tubes, bowls, and quasi-planar triangular networks), and exhibits strong aromaticity. In 2018 and 2020, Li et al. presented two new axially chiral members, cage-like B_34_, $B_{35}^{+}\;$ and seashell-like $B_{31}^{+}$, B_32_, to the borospheren family [15,16], and revealed the universal bonding pattern of σ + π double delocalization in this type of borospheren.

Because of the important application of boron in the nuclear industry and the peculiar physical and chemical properties of actinides, the interaction between actinides and borophene is also a research hotspot. Wang et al. reported a unique actinide-encapsulated U@B_40_ cage structure using density functional theory (DFT) calculations, and indicated that U@B_40_ exhibits a 32-electron closed-shell configuration [17]. Shi et al. explored a series of actinide borospherenes AnB_n_ (An=U, Th; n = 36, 38, and 40) [18]. Their results indicate that doping with the right actinides may stabilize various boroballenes and open up an avenue for boroballene modification and functionalization. In 2020, Du et al. investigated M@B_36_ (M = Ti, Zr, Hf, Ce, Th, Pa^+^, U^2+^, Np^3+^, and Pu^4+^), which all meet the 32-electron principle [19]. 

In this work, a series of actinide metalloborospherenes, An&B_m_ and An@B_n_ (An=U, Np, Pu; m = 28, 32, 34, 36, 38, 40; n = 36, 38, 40), have been examined systematically. Electronic structures, bonding characteristics, charge transfers, and IR and UV-vis spectra were predicted. The solvation effect in an aqueous solution is discussed in all processes of the research. In addition, we systematically compared the properties of the exohedral and endohedral structures in order to obtain a relatively comprehensive understanding of actinide metalloborospherenes.

## 2. Results

### 2.1. Exohedral Actinide Borospherenes

The structure of pure boron clusters has been well studied, and the structural information has been vividly described. B_28_ and B_32_ are seashell-like borospherene cages with *C*_2_ symmetry, B_34_ is aromatic and cage-like with *C*_2_ symmetry. The planar B_36_ can be transformed into a cage-like structure with the doping of actinide atoms. B_40_ is a fullerene-like cage with *D*_2*d*_ symmetry.

The electrostatic potential (ESP) on the molecular vdW surface analysis of borospherenes was depicted to determine the adsorption sites of actinides, and the corresponding ESP-mapped molecular surfaces are shown in the Appendix A.

According to the ESP, actinide was decorated on the proper site (the lowest PES value) of borospherenes for the geometry optimization. In addition, different spin multiplicities were considered in the structure optimization process, and the ground-state structure was examined by the TDDFT method. The results of TDDFT verification show that the excitation energies of the obtained structure are positive, indicating that these electronic structures are more stable than the excited state. The corresponding geometry coordinates of An&B_m_ and An@B_n_ (An=U, Np, Pu; m = 28, 32, 34, 36, 38, 40; n = 36, 38, 40) are listed in Appendix A.

The optimized molecular structures of the An&B_m_ complexes are shown in Figure 1, and the average bond length of An-B are listed in Table 1. The An-B average bond lengths for the identical actinides exhibit a tendency of first falling and then rising, and boron clusters of the same size indicate an overall rise from U to Pu, which may be a result of the atomic size for actinides. The An-B chemical bond, on the other hand, is lengthened in the solution instance.

Bonding energies were used to evaluate the interaction strength of An-B and the results are plotted in Figure 2. For boron clusters of the same atomic number, the binding energy of U-B is the strongest, and the variation in the amount of boron atoms is minor. The connection of Pu-B, on the contrary, is the weakest. The change is larger as the number of boron atoms increases, and the fluctuation range is 4–7 eV.

The representative optimized structure of An&B_m_ is shown in Figure 3. As shown, for the geometry of An&B_36_, where An is just above the center of the hexagonal hole, and B_36_ has little deformation.

To gain further insight into the interaction features of An&B_m_, we performed an electron density topological analysis using an electron local density function (ELF) and the quantum theory of atoms in molecules (QTAIM) method. The topological parameters of the An–B bond critical points (BCPs) of AnB_36_ as the representative metalloborospherenes are listed in Table 2, and the corresponding parameters of other complexes are collected in Appendix A.

Previous studies [20] of actinide-containing systems have proven that the energy density proposed by Cremer and Kraka [21] can be used as the criterion to correctly explain the nature of chemical bonds. The more negative the *H*(*r*) value, the more obvious the covalent character. For typical ionic bonds, the *−V*(*r*)/*G*(*r*) < 1, and for classical covalent bonds, the *−V*(*r*)/*G*(*r*) > 2.

As shown in Table 2, two-bond critical-point An-B BCPs indicate the existence of An-B bond interactions in An&B_36_. The negative *H*(*r*) as well as *−V*(*r*)/*G*(*r*) between 1 and 2 indicating the An–B bonds are partial covalent interactions. Moreover, from U to Pu, the *H*(*r*) value is gradually increasing, and the ratio *−V*(*r*)/*G*(*r*) is gradually decreasing and tends to be 1, indicating that the covalent character is gradually weakening.

For other complexes of An&B_m_, except for the number of bond critical points, the properties of chemical bonds and the changing rules of topological parameters are consistent with An&B_36_, as reflected in Appendix A.

Selected bond lengths, the fuzzy bond order (FBO), and the Hirshfeld charge of the AnB_36_ complex in a vacuum are listed in Table 3. The ground states of UB_36_, NpB_36_, and PuB_36_ are triplet, quartet and quintet, respectively. From U to Pu, the fuzzy bond order value becomes smaller and the bond becomes longer, indicating that the covalent bond becomes weaker. Here, the actinide atoms possess a huge electric charge, which in turn leads to a large dipole moment shown in Appendix A. To explain this charge distribution, we created a map of the electron density differences to accurately and comprehensively describe the transfer process. As displayed in Figure 4, the shapes of their density-difference maps are similar. It can be seen that there are positive-valued regions between An-B, indicating that the formation of covalent bonds is accompanied by the accumulation of electron density between the bonding atoms.

DOS analysis was used to investigate the orbital characteristics in depth. The total, partial and overlap population density of the state (TDOS, PDOS and OPDOS) curves of Np&B_36_ are plotted in Figure 5, and the corresponding images of the other An&B_36_ examples are plotted in Appendix A. Fragment 1 is defined as An atomic orbits, and Fragment 2 is defined as B_36_ orbits. Taking Np&B_36_ as an example, it can be seen that below the highest occupied orbital, B36 contributes to almost all of the density of states. Moreover, there is an interaction between B_36_ and Np atoms, which can be explained by the OPDOS value being greater than zero. This is consistent with the conclusion of the previous QTAIM analysis.

Figure 6 presents the IR spectra of U&B_36_, Np&B_36_ and Pu&B_36_ complexes and the vibrational modes of the corresponding peaks. The vibrations corresponding to the high-frequency peaks are mainly contributed by B_36_, and the peak frequencies gradually decrease from U to Pu. Two modes at 166.5 and 235.9 cm^−1^ were assigned to the contraction vibration of the U atom and B_36_. In contrast, the vibration peaks of Np&B_36_ and Pu&B_36_ in the low-frequency region are increased, but there is no contraction vibration of the An atom and B, such as U&B_36_. More IR and UV-vis spectras of An&B*_m_* are available in Appendix A and IR and UV-vis spectras of An@B*_n_* shown in Appendix A.

### 2.2. Actinide Endohedral Borospherenes

The predicted low-lying endohedral structures of the studied An@B_n_ at PBE level are depicted in Figure 7, and the corresponding structural parameters are listed in Appendix A. Except for U@B_38_ and U@B_40_, the multiplicity of the An@B_n_ structure remains unchanged compared with An&B_n_. Both ground states of U@B_38_ and U@B_40_ are singlet. In addition, U@B_36_ shows the property of a 32-electron closed-shell singlet and the shells of *s*, *p*, *d* and *f* are filled. The minimum frequencies of U@B_40_, Np@B_40_ and Pu@B_40_ are 26.92, 54.01 and 54.77 cm^−1^, respectively.

As depicted in Figure 8, the binding energy differences of An@B_n_ in vacuum and in aqueous solution were less than 0.5 eV, except for Np@B_38_ and Pu@B_40_. The binding energy of the endohedral structure is double that of the exohedral structure for the same chemical formula. Taking UB_36_ as an example, the binding energies of U&B_36_ and U@B_36_ are 16.4 and 7.7 eV, respectively.

We evaluated the effect of solvation effects on the dipole moment, energy and structure, and the results are presented in Appendix A. As can be seen, in An&B_m_ or An@B_n_, the value of the dipole moment is larger in the aqueous solution than in vacuum, but the rotational constants and single-point energy are almost unchanged. Compared with An@B_n_, the dipole moment of An&B_m_ is significantly increased, the rotational constants are small, and the single-point energy is also almost unchanged. 

The QTAIM and ELF analyses results of U@B_n_ are plotted in Figure 9 and Appendix A. As can be seen, there is an obvious covalent interaction between the boron atoms, which is manifested in the existence of the critical point of the bond and the disynaptic valence basin of the ELF. In contrast, the prominent closed-shell interactions between U atoms and boron atoms are exhibited, and they exhibit strong multicenter An-B bond characteristics, manifested in the existence of the cage critical points.

To obtain the depth characteristics of the UV-vis spectrum, the charge-transfer spectrum (CTS) [22] of An@B_n_ has been plotted. The CTS of representative structures U@B_36_ are presented in Figure 10 and the rest are presented in Appendix A.

As shown in Figure 10, the redistribution of B_36_ has the largest contribution to UV–vis, and shows the obvious characteristics of electron transfer from B_36_ to U. The optical absorption of the U@B_36_ is mostly induced by the electronic excitation of the B_36_, and the absorption around 406 nm comes to a certain extent from the electron transfer from B_36_ to U, which is triggered when photons are absorbed. The figure also shows that the transition between the atomic orbitals of U and the U→B_36_ electron transfer contribute less to the excitation of optically active electrons, mainly because the valence electron orbital of the U atom is full and cannot be transferred.

We took into account the spin–orbit coupling (SOC) effects on orbital energies of actinide borospherenes. Our previous research [20] and the U@B_40_ results of other research groups [17] have shown that spin–orbit coupling has little effect on the structure and other properties of actinide complexes. Here, we calculated the *f*-orbital populations of the actinides and the complexes, and the data are shown in Appendix A. It can be seen that the *f*-orbital population is consistent, indicating that the calculation results are credible in the case of ignoring the SOC.

## 3. Computational Methods

Geometry optimization and frequency calculation were performed with the PBE [23] method without symmetry constraints using an ORCA 4.2.1 package [24,25]. Zero-order regular approximation (ZORA) [26] was employed to consider the scalar relativistic effect of the actinide atom. For the basis set, ZORA-SARC [27,28] and ZORA-def2-TZVPP were applied for the actinides and boron, respectively. The singlet-point energies were calculated employing the PWPB95 [29] method, in which D3 [30] stood for the Grimme’s atom pairwise dispersion correction, and BJ for Becke–Johnson damping. Time-dependent density functional theory (TDDFT) was calculated in a PBE0 [31]-def2/TZVPP level with 100 excited states calculated. The binding nature has been deeply investigated by the Multiwfn package [32]. The bonding characteristics were studied with several reliable methods, such as QTAIM [33], fuzzy bond order (FBO) [34,35], total, partial and overlap population density of state (TDOS, PDOS and OPDOS, respectively) [36,37], electron local density function (ELF) [38], and Voronoi deformation density (VDD) atomic charge population [39]. The solvent effect of the aqueous solution on the UV–vis spectra and binding energies was considered by using the conductor-like polarizable continuum model (CPCM) [40].

## 4. Conclusions

In conclusion, a series of exohedral actinides and endohedral borospherenes (An&B_m_ and An@B_n_ (An=U, Np, Pu; m = 28, 32, 34, 36, 38, 40; n = 36, 38, 40)) were investigated. The electronic structures, orbital characteristics and spectral information of the systems were systematically compared in order to obtain a more comprehensive understanding of the actinide borospherenes. The current results demonstrate that the endohedral structures are more stable than the exohedral structures, and the stability presents a trend of UB_m_ > NpB_m_ > PuB_m_, except for PuB_40_. Borospherenes contribute to the majority of the DOS and UV–vis in such systems, and there is an electron transfer from borospherenes to actinides. Furthermore, the valence electron orbital of an An atom is more readily occupied for specific unique embedded structures, and can be qualified as a 32-electron system. The findings show that doping actinide metal atoms into borospherenes may be used to chemically change and functionalize them, improving their stability and changing their surface reactivity. In view of the diversity of boroballenes and actinides, we will further study the interaction characteristics of actinide oxides with various boroballenes. The ultimate goal of these studies is to design and synthesize corresponding actinide-containing materials. Therefore, the connection and transition from composite properties to material properties will also be the focus of follow-up research.

## Figures and Tables

**Figure 1 molecules-27-06047-f001:**
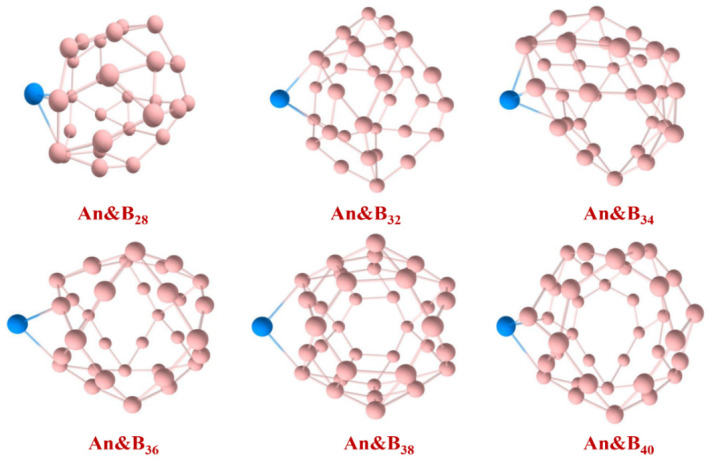
Optimized structures of An&B_m_ at the PBE-ZORA/def2-TZVPP-SARC level. The adsorption sites of An atoms were obtained by geometric optimization on the basis of the reactive sites.

**Figure 2 molecules-27-06047-f002:**
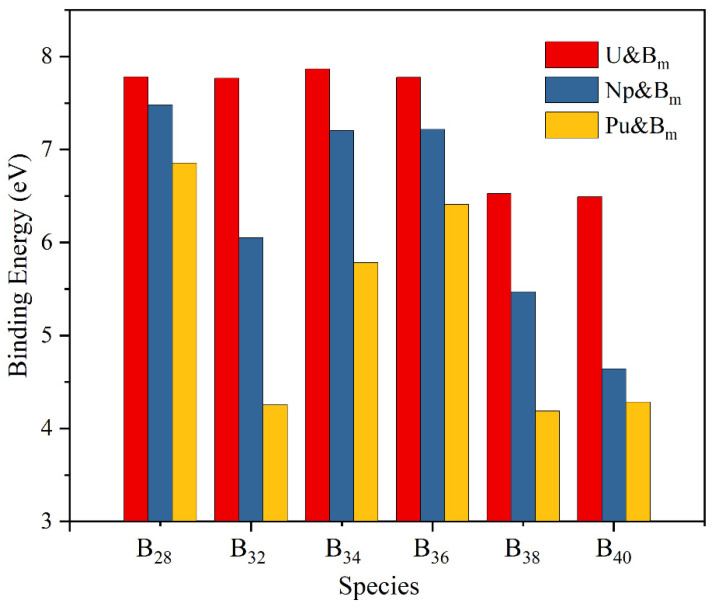
The binding energy (eV) of ground-state An&B_m_ (An=U, Np, Pu; m = 28, 32, 34, 36, 38, 40) in PWPB95-ZORA/def2-TZVPP-SARC level.

**Figure 3 molecules-27-06047-f003:**
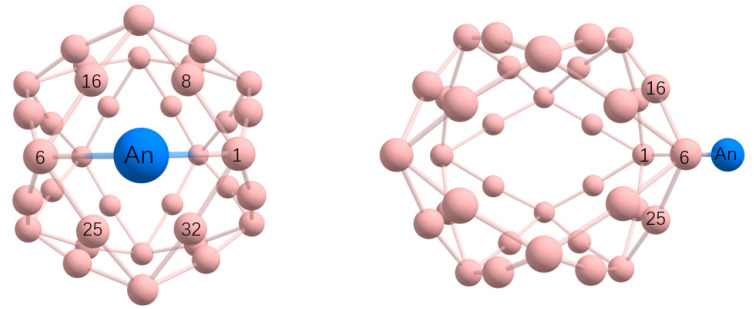
Optimized geometries of An&B_36_ (An=U, Np, Pu) at the PBE-ZORA/def2-TZVPP-SARC level. The corresponding geometry coordinates are listed in Appendix A.

**Figure 4 molecules-27-06047-f004:**
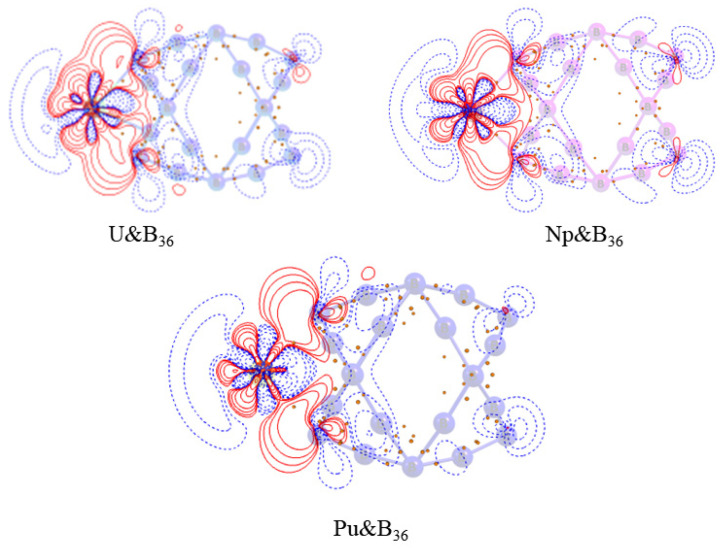
Contour plots of the electron density difference between An and B_x_ fragments. Solid lines (red) represent regions where electron density increases, the dotted lines (blue) represent the region where the electron density decreases, and the green dots represent the BCPs.

**Figure 5 molecules-27-06047-f005:**
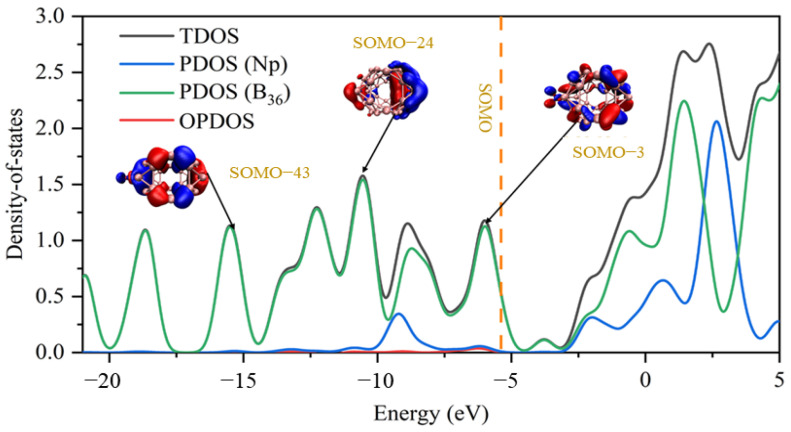
The TDOS, PDOS and OPDOS curves of Np&B_36_ at the PBE-ZORA/def2-TZVPP-SARC level. TDOS, PDOS and OPDOS are broadened to Gaussian curves and the full width at half-maximum (FWHM) is 0.8 eV.

**Figure 6 molecules-27-06047-f006:**
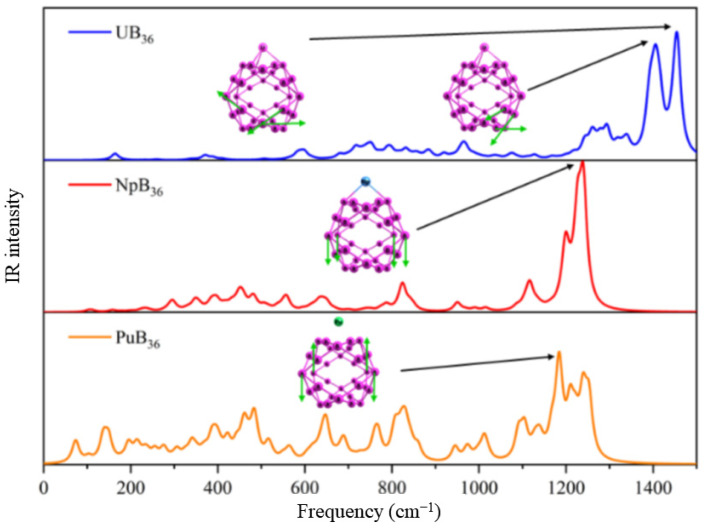
IR spectra of An&B_36_ (An=U, Np, and Pu) cluster at the PBE-ZORA/def2-TZVPP-SARC level, plotted by broadening discrete lines with Lorentzian function setting full width at half-maximum (FWHM) as 20 cm^−1^.

**Figure 7 molecules-27-06047-f007:**
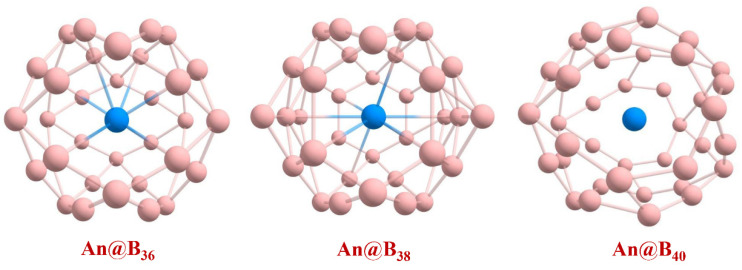
Optimized geometries of An@B_n_ at the PBE-ZORA/def2-SVP-SARC level.

**Figure 8 molecules-27-06047-f008:**
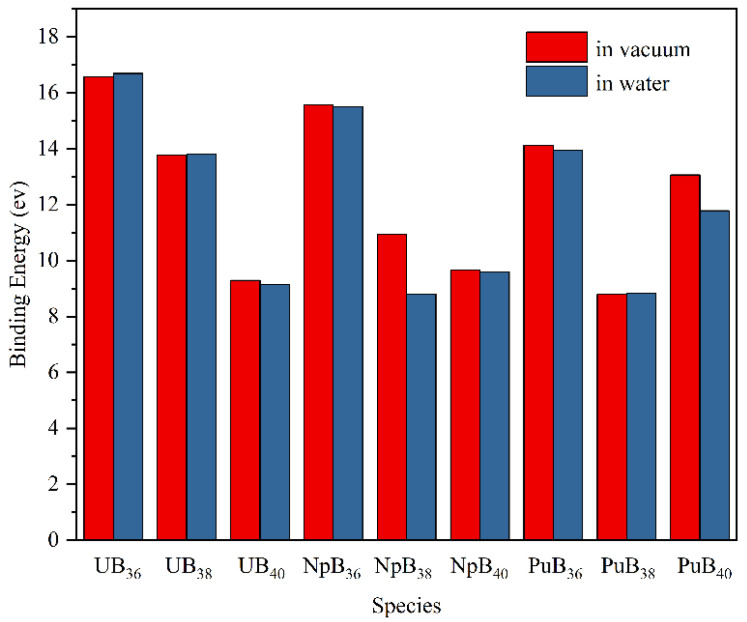
Binding energies of An@B_n_ in vacuum and in aqueous solution at the PWPB95-ZORA/def2-TZVPP-SARC level.

**Figure 9 molecules-27-06047-f009:**
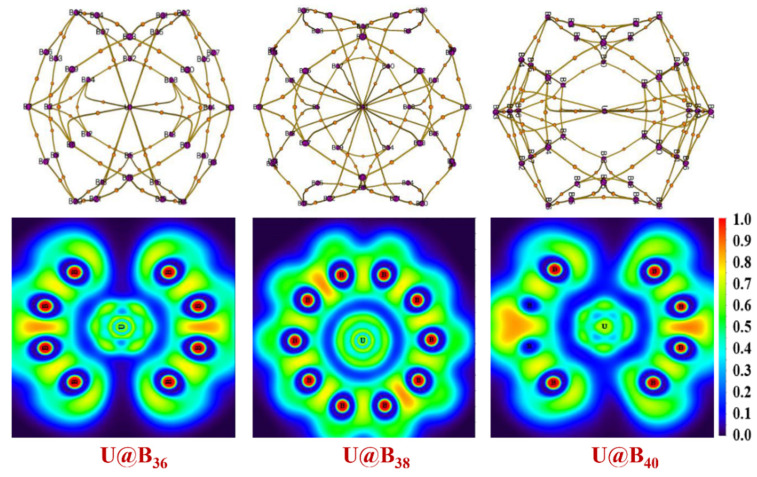
The critical-point molecular graph and two-dimensional color-filled plane maps of ELF for U@B_36_, U@B_38_ and U@B_40_. Red points represent bond critical points, lines represent bond paths.

**Figure 10 molecules-27-06047-f010:**
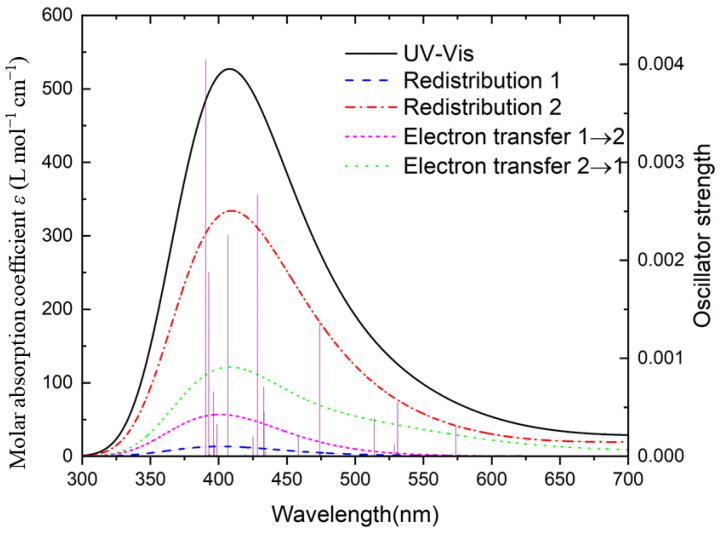
Charge-transfer spectrum of U@B_36_ at PBE0-ZORA/def2-TZVPP-SARC level. The FWHM is 0.67 eV with Gaussian curves. The system is divided into two fragments, 1 for U and 2 for B_36_. The blue (dashed) line represents the redistribution of fragment U, and red (dash-dotted) line represents the redistribution of fragment B_36_. The magenta (short dashed) line and the green (dotted) line denote MLCT (metal–ligand charge transfer) and LMCT (ligand-to-metal charge transfer), respectively.

**Table 1 molecules-27-06047-t001:** Average bond length $\bar{r}\left(Å\right)$ of An and adjacent B atom in An&B_m_ in vacuum and in aqueous solution using the C-PCM model (in parentheses) at PBE-ZORA/def2-TZVPP-SARC level.

$\bar{r}\left(Å\right)$	B_28_	B_32_	B_34_	B_36_	B_38_	B_40_
**U**	2.57	2.55	2.54	2.49	2.52	2.57
	(2.62)	(2.59)	(2.58)	(2.56)	(2.56)	(2.59)
**Np**	2.59	2.55	2.55	2.52	2.52	2.57
	(2.64)	(2.62)	(2.60)	(2.59)	(2.58)	(2.61)
**Pu**	2.60	2.57	2.56	2.54	2.53	2.59
	(2.64)	(2.63)	(2.61)	(2.62)	(2.59)	(2.65)

**Table 2 molecules-27-06047-t002:** Topological parameters for the An–B bond critical points (BCPs) of the AnB_36_ clusters at PBE-ZORA/def2-TZVPP level *.

Species	Bond	*ρ*(*r*)	$\nabla^{2}\rho \left(r\right)$	*G*(*r*)	*V*(*r*)	*H*(*r*)	*−V*(*r*)/*G*(*r*)	ELF
**UB_36_**	U-B_1_	0.084	0.090	0.052	−0.082	−0.030	1.569	0.446
	U-B_6_	0.084	0.092	0.053	−0.082	−0.029	1.560	0.440
**NpB_36_**	Np-B_1_	0.077	0.096	0.049	−0.073	−0.025	1.505	0.406
	Np-B_6_	0.077	0.096	0.049	−0.073	−0.025	1.505	0.405
**PuB_36_**	Pu-B_1_	0.073	0.103	0.048	−0.070	−0.022	1.461	0.375
	Pu-B_6_	0.073	0.103	0.048	−0.070	−0.022	1.461	0.375

* Parameters are density of all electrons *ρ*(*r*), Laplacian of electron density ∇^2^*ρ*(*r*), Lagrangian kinetic energy *G*(*r*), potential energy density *V*(*r*), total energy density *H*(*r*) [*H*(*r*) *= G*(*r*) *+ V*(*r*)] and ELF.

**Table 3 molecules-27-06047-t003:** Selected bond lengths *r*(Å), fuzzy bond order (FBO), Hirshfeld charge of An&B_36_ complex in vacuum.

Species	2S + 1	Bond	*r*(Å)	FBO	Hirshfeld
**U&B_36_**	3	U-B_1_	2.36	1.04	−4.28739 (U)
		U-B_6_	2.36	1.06	
**Np&B_36_**	4	Np-B_1_	2.39	0.97	−4.34314 (Np)
		Np-B_6_	2.39	0.97	
**Pu&B_36_**	5	Pu-B_1_	2.40	0.97	−4.35974 (Pu)
		Pu-B_6_	2.40	0.97	

## Data Availability

Not applicable.

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
