# Peer review of "A Systemic Insight into Exohedral Actinides and Endohedral Borospherenes: An&Bm and An@Bn (An=U, Np, Pu; m = 28, 32, 34, 36, 38, 40; n = 36, 38, 40)"

_molecules, 2022, doi:10.3390/molecules27186047_

Round 1

Reviewer 1 Report

This is a DFT study of exo- and endohedral complexes between selected boron clusters and actinide atoms. The study is well motivated, and the results are clearly presented. Some interesting trends are reported, which I think are worthy of publication. I have a number of methodological concerns, however, which should be addressed in a revised version.

My main concern is the neglect of spin-orbit coupling (SOC) for the heavy actinides. For the f(n) systems (n = 6 – 8) SOC should be massive. If, as has for instance been shown for the actinyl series, [AnO2](n+), the extent of SOC depends on the number of f-electrons, then relative energies between structures with different electron configurations are dubious at best. Arguably this may apply to the binding energies reported in Fig. 2, where population of the f-orbitals on the metal will have changed upon electron transfer to the cage. In that context, the electron configurations of the free actinide atoms used in the calculations must be disclosed and the expected accuracy of the results discussed in the light of the problems of DFT to correctly reproduce atomic ground states of d-block transition metal atoms already.

The same argument would also apply to the binding energies in Fig. 8. However, if the extent of charge transfer from metal to cage (and, thus, the f-orbital population on the metal) would be similar for exo- and endohedral species, the relative energies between the two could be more reliable. These energies (along with the binding energies) should be given in tabular form for all species (can be deposited in the ESI). In fact, many of these energies will be available at multiple levels (from PBE optimizations and from the PBE0, PWPB95 and PWPB95-D3 single points), all should be reported (again in the ESI) to gauge the dependence on the functional.

Apparently a PCM has been used to evaluate solvation effects on the binding energies for the endohedral complexes (Fig. 8), but not for the exohedral ones (Fig. 2). This is surprising given that the latter should have much larger dipole moments than the former (which should actually be reported and discussed), and should therefore be stabilized more strongly in a polar solvent (or environment). In that context, the use of water as model solvent should be justified – surely the authors don't expect systems like that to be stable in aqueous solution?

In cases where different multiplicities have been trialled to find the electronic ground state, the relative energies of all structures should be given as ESI. And again, the metal f-orbital populations need to be reported for all these states, because in the absence of SOC corrections, these energies (and, thus, the identification of the electronic ground state) can only be trusted if that f-population remains the same.

In summary, with more documentation, a much more critical discussion of SOC effects and additional PCM calculations, the paper may be suitable for acceptance, but it should be reviewed again.

Two final quibbles: Normally I don't like the presentation of DOSs (a concept for infinite periodic solids) for discrete molecules - why not discuss the MOs in terms of their energy levels and AO composition? Lasty, an important typo on p. 8, l.180: the unit should be eV, I assume, not cm-1.

Author Response

Thanks for your comments on our paper. We have revised the manuscript carefully according to your suggestions.We here submit the revised manuscript as well as a list of detailed changes (point-by-point response to the reviewer comments) in the attachment. Please see the attachment.

Reviewer 2 Report

This manuscript is interesting for both boron and actinide chemistry and could be published in Molecules. The topic is definitely original. A series of actinide exohedral borospherenes An&Bm and endohedral borospherenes An@Bn (An=U, Np, Pu; m=28, 32, 34, 36, 38, 40; n= 36, 38, 40) have been characterized by density functional theory calculations. The electronic structures, chemical bond topological properties and spectra have been systematically investigated. It was found that An@Bn is more stable than An&Bn in terms of structure and energy, and UB36 in aqueous solution is the most stable molecular in this research. The IR and UV-vis spectra of An&Bm and An@Bn are computationally predicted to facilitate its further experimental investigations. Charge transfer spectroscopy decomposes the total UV-Vis absorption curve into the contributions of different excitation features, allowing insight into what form of electronic excitation the UV-Vis absorption peak is from the perspective of charge transfer between the An atoms and borospherenes. 

The introduction provide very sufficient background. The research methodology is adequate and modern. The results are clearly presented. The amount of data is large. The conclusions supported by the data. The manuscript good illustrated and interesting to read. English language and style are fine, and may be very minor polishing from native speaker is recommended. I have also couple of minor suggestions:

- Following relevant references focused on theoretical studies of boron clusters should be briefly cited in introduction: Int. J. Mol. Sci. 2022, V. 23. P. 4190.; Polyhedron 2022, V. 211. P. 115559.

- Some more detailed perspectives regarding the future research could be formulated in conclusions section.

Overall, this nice manuscript could be accepted for publication after minor revisions.

Author Response

Thanks for your comments on our paper. We have revised the manuscript carefully according to your suggestions. We here submit the revised manuscript as well as a list of detailed changes (point-by-point response to the reviewer comments) in the attachment. Please see the attachment.

Round 2

Reviewer 1 Report

As far as I can see, most of my comments have been addressed in the cover letter, though not all in the revision. That orbital energies do not seem to be drastically affected by spin-orbit coupling (at least for one representative complex) is reassuring, although it is the final total energies that would be more relevant in that context. In any event, potential shortcomings due to the neglect of spin-orbit coupling should be conceded in the paper and a brief discussion (for instance citing the paper mentioned in the cover letter) should be added to the ms. In that discussion it would also make sense to refer to the f-orbital populations in Tabs. S7-S9, conceding that these seem to show some variability with the chemical environment and that for more accurate relative energies (e.g. actinide binding energies) follow-up studies with explicit SOC included might be warranted. As a note on the side, apparently 4f and 5f populations are summed up in Tabs. S7-S9, it is mainly the 5f one that would be of interest here.

One reply that raises an additional question is that on the atomic charges in An&B36 (Tab. S4 in the ESI): Apparently here the actinide atoms are predicted to carry huge negative charges (exceeding -4), leading to very large dipole moments (cf. Tab. S11) – this would seem strange with an electropositive metal and an electron-deficient cage. As expected this leads to noticeable changes in the single-point energies between gas phase and solution as documented in that table. This is probably not hugely important as the endohedral species are computed to be much more stable throughout, but brief comment on this finding would still seem in order.

After these further minor revisions, I am happy to recommend acceptance.

Author Response

Thank you again for your comment on our manuscript. We have revised the manuscript carefully according to your suggestions, detailed responses are as follows: (in order to minimize the amount of time that the reviewers distinguish their comments and our reply, the reviewer’s comments are marked by the blue, the modifications are marked in red.) Please see the attachment.
